# The Probiotic *Streptococcus salivarius* M18 Increases Plasma Nitrite but Does Not Alter Blood Pressure: A Pilot Randomised Controlled Trial

Mia C. Burleigh [1,*], Bob T. Rosier [2], Annabel Simpson [1], Nicholas Sculthorpe [1], Fiona Henriquez [1] and Chris Easton [1]

[1] Sport and Physical Activity Research Institute, School of Health and Life Sciences, University of the West of Scotland, Glasgow G72 0LH, UK; annabel.simpson@uws.ac.uk (A.S.); nicholas.sculthorpe@uws.ac.uk (N.S.); chris.easton@uws.ac.uk (C.E.)

[2] Centre for Advanced Research in Public Health, Department of Health and Genomics, FISABIO Foundation, 46020 Valencia, Spain

[*] Correspondence: mia.burleigh@uws.ac.uk

**Abstract:** Some species of oral bacteria can reduce dietary nitrate to nitrite, which can later be converted to nitric oxide in the nitrate—nitrite—nitic oxide pathway. Increasing nitric oxide availability can reduce blood pressure (BP) and improve exercise performance. *Streptococcus salivarius* M18 (*Streptococcus salivarius* M18) is a bacteriocin-producing probiotic that is known to improve oral health by inhibiting pathogenic oral bacteria. However, it is presently unclear whether probiotic-induced alterations to the oral microbiome will influence circulating levels of nitric oxide metabolites and BP. Purpose: To determine the effects of *Streptococcus salivarius* M18 supplementation on plasma and salivary nitrate and nitrite levels and BP. Methods: Ten healthy males (age 32 ± 8 y, body mass 88.2 ± 15.1 kg) completed 2 × 14-day supplementation phases in a randomized order at least 14 days apart. In one phase, participants consumed *Streptococcus salivarius* M18 probiotic lozenges (2.5 billion colony-forming units/dose) once per day, and in the other, they ingested water (placebo). The abundance of bacteria on the tongue was assessed via Illumina 16S rRNA gene sequencing, unstimulated saliva, and venous blood samples were collected, and BP was measured pre and post each phase. Saliva and plasma were analysed for nitrate and nitrite using chemiluminescence, and pH was measured in saliva. The change in each outcome from pre- to post-supplementation was compared between phases using repeated measures ANOVA. Results: Plasma nitrite increased from baseline following probiotic supplementation (from 173 ± 39 to 223 ± 63 nM, *p* = 0.003, 95% CI 192–250 nM). In comparison, there was no change in the placebo phase or between baselines (all *p* > 0.05). The abundance of nitrite-producing bacteria was not altered, salivary nitric oxide metabolites and pH did not change, and the increase in plasma nitrite did not result in reductions in BP (all *p* > 0.05). Conclusions: Supplementation with *Streptococcus salivarius* M18 increased plasma nitrite, a key marker of NO availability. Despite this, *Streptococcus salivarius* M18 did not lower BP in these healthy normotensive participants. Additionally, the increase in plasma nitrite was not associated with abundance changes in bacteria thought important to NO generation. Further research is required to determine the mechanism behind the increase in plasma nitrite and the potential therapeutic and ergogenic benefits of *Streptococcus salivarius* M18 supplementation.

**Keywords:** nitric oxide 1; nitrate 2; nitrite 3; probiotic; *Streptococcus salivarius* M18; blood pressure; oral health; cardiovascular health

## 1. Introduction

Probiotics are commonly used as therapeutic agents to promote gut health [1] and have been shown to act on systemic health via a range of mechanisms. Strains such as *Lactobacillus reuteri* NCIMB 30242, commercially named Cardioviva, have demonstrated improvements

in blood lipid chemistry following administration [2,3]. Furthermore, some probiotics have been shown to reduce systolic and diastolic blood pressure in pre-hypertensive and hypertensive patients [4]. *Lactobacillus plantarum* 299 v protected rats against myocardial infarction via reduction of serum leptin [5], and a recent review has highlighted the potential of gut probiotics to improve cardiovascular health via reduction of oxidative stress [6].

More recently, targeted strains have been developed to support oral health [7]. *Streptococcus salivarius* M18 is an oral probiotic developed from a naturally occurring strain of the human oral cavity [8] found mostly on the tongue surface [9]. This probiotic has been investigated and found to be potentially effective in the treatment of pharyngitis, dental caries, periodontitis, and halitosis [10]. Improvements in oral health following *S. salivarius* M18 are proposed to result from the crowding out of pathogenic species by adhesion to the gingival tissue, tongue surface, or hard surfaces of the teeth. Alternatively, this strain may change the metabolic activity of resident flora by producing bacteriocin-like inhibitory substances (BLIS) such as dextranase and urease. These mechanisms appear to combine to reduce dental plaque accumulation and acidification [11]. *S. salivarius* M18 is also known to improve indices of the inflammatory conditions of gingivitis and periodontitis by balancing anti-inflammatory and pro-inflammatory cytokines [12]. Some strains of *S. salivarius* also produce nitrite [13], which could result from the reduction of nitrate or other metabolic pathways [14]. Nitrite can be further reduced by oral bacteria to nitric oxide—a molecule with anti-microbial properties and a biofilm dispersal signal for some bacteria [15].

While oral probiotics have shown positive outcomes in the oral cavity, it is, as yet, unknown what effects introducing a new bacterial strain to the existing microbial community may have on the nitrate—nitrite—nitric oxide pathway. Recent evidence shows that certain species of the resident oral microbiota are critical to the reduction of nitrate from the diet [16,17] and that their depletion results in the loss of the cardioprotective benefits of dietary nitrate [16,18,19]. Unlike gut probiotics, oral probiotics do usually not survive gastric passage and are, therefore, not thought to provoke systemic effects [20]. However, given the interconnectedness of oral health status and cardiovascular risk [21] and recent evidence confirming the importance of the oral microbiome to the nitrate—nitrite—nitric oxide pathway [17,22–24], it would be premature to rule out the possibility of systemic effects resulting from probiotics interacting with the oral nitrate-reducing community.

Therefore, our primary objective was to investigate the effects of the oral probiotic *Streptococcus salivaris* M18 on the structure of bacterial communities previously associated with nitric oxide reduction on the dorsal surface of the tongue, metabolites of nitric oxide in the blood and saliva, and blood pressure measurements in a group of healthy males.

## 2. Materials and Methods

### 2.1. Ethical Considerations

Ethical approval was given by the School of Health and Life Sciences Ethics Committee at the University of the West of Scotland—approval number 2018-5000-3633—date May 2018. All procedures described were conducted in accordance with the Declaration of Helsinki 1974 and its later amendments.

### 2.2. Participants

Ten healthy males (age 32 ± 8 years, stature 178 ± 6 cm, and body mass 88.2 ± 15.1 kg) volunteered and provided written informed consent prior to participating in this pilot study. Participants were excluded from taking part if they reported use of any use of antibacterial mouthwash or antibiotics for at least six months prior to study commencement. They were free from non-prescription medication, including those known to interfere with stomach acid production, and were not taking any prescribed medication. Health status was confirmed by the completion of a medical questionnaire (Supplementary File S1), and the World Health Organisation's oral health questionnaire for adults [25] was used to ascertain oral health status. Inclusion/exclusion criteria can be found in Supplementary

File S2. Trials were performed at the University of the West of Scotland (55.78° N latitude) between September and December.

### 2.3. Experimental Design

Participants were required to attend the laboratory on four separate occasions for this placebo-controlled, double-blind, randomised, crossover study. The study comprised two separate fourteen-day dietary supplementation phases, each preceded by a baseline trial (day 0) and completed with a post-supplementation trial (day 15). In one phase, participants consumed *Streptococcus salivarius* M18 probiotic lozenges (2.5 billion colony-forming units/dose) once per day, and in the other, they ingested 40 mL of water (placebo). There was a four-week washout period between phases. The CONSORT reporting guidelines were used (Supplementary File S3).

### 2.4. Blinding

Participants were told that in one phase, they would receive a liquid probiotic (water), and in the other, they would receive a lozenge probiotic. Groups were assigned randomly by the lead researcher using https://www.random.org (accessed on 1 June 2018). Following randomisation, participants collected their supplements in opaque envelopes. They were asked not to reveal in which order they received the supplements. Following the completion of data collection, and before it was revealed that the liquid probiotic was a placebo, the participants were asked whether they preferred the liquid or the lozenge probiotic. Seven out of the ten participants said they preferred the liquid probiotic for ease of use. One preferred the lozenge for reasons of taste; the remainder had no preference.

### 2.5. Consumption of the Supplements

Participants were asked to consume their allocated supplement in the evening between 7 pm and 10 pm. The first consumption time was recorded and repeated throughout the study. Visits timings were arranged so that the last dose was consumed 12 h before arrival in the laboratory. Participants were asked not to chew the lozenge but to allow it to dissolve on the tongue. They were also asked not to brush their teeth, eat, or drink within thirty minutes of probiotic consumption. The instructions for consumption of the placebo were identical, with the exception that they were asked to hold the liquid in their mouth for 20 s before swallowing.

### 2.6. Procedures

Participants were briefed on procedures and provided with a food diary in which they recorded all food and drink consumed seven days prior to the first trial and throughout the supplementation period. This diary was used to replicate the diet in the week preceding and during the second supplementation phase. Participants arrived at the laboratory on the morning of each trial after a 12 h overnight fast. Upon arrival, they were given 500 mL of water. The water was consumed 1 h before the first measurements were collected. Strenuous exercise was avoided for 24 h, and caffeine for 12 h before each trial. Participants were requested not to brush their teeth and tongue or chew gum on the morning of each trial. Mouthwash use was not permitted throughout the study, and participants were asked to report any changes in health status. A checklist was completed on each visit to check compliance with the pre-trial instructions.

Body mass was recorded at the beginning of each visit using conventional methods. Following this, participants lay supine for the remainder of the experiment. The posterior dorsal surface of the tongue was swabbed for 1 min with a sterile Hydraflock swab (Puritan HydraFlock Swabs, Puritan Diagnostics LLC, Guilford, ME, USA). This area of the tongue is known to harbour nitrate-reducing bacteria and is where the majority of nitrate-reduction activity occurs [26]. The swabs were stored in transport tubes containing 0.85 mL of buffered sterile saline and 0.15 mL of glycerol and stored at −80 °C.

No further measurements were collected for 30 min to ensure plasma nitrite had stabilised following the change in body posture [27]. Systolic BP (SBP) and diastolic BP (DBP) were recorded in triplicate using an automated device (Orman M6, Intelli-Sense. Hoofdorp, The Netherlands). Mean arterial pressure (MAP) was calculated using the following equation:

$$MAP = (2 \times DBP + SBP)/3 \tag{1}$$

Venous blood was collected via venepuncture of the forearm in 4 mL aliquots in vacutainer tubes containing ethylenediaminetetraacetic acid (BD vacutainer K2E 7.2 mg, Plymouth, UK). Samples of whole blood were immediately centrifuged for 10 min at 4000 rpm at 4 °C (Harrier 18/80, Henderson Biomedical, London, UK) following collection. Samples of unstimulated saliva were collected directly into a mini centrifuge tube and centrifuged at 4000 rpm. Following centrifugation, the samples of plasma and saliva were immediately stored at −80 °C for later analysis of nitrate and nitrite content via ozone-based chemiluminescence as previously described [17].

### 2.7. 16S Metagenomic Sequencing

DNA was extracted from the tongue swabs using the MasterPureTM Complete DNA and RNA Purification Kit (Epicentre, Madison, WI, USA) following the manufacturer's instructions. Isolated DNA was shipped to the Oral Microbiome Laboratory, Genomics and Health department, FISABIO Foundation, Valencia, Spain, where an Illumina amplicon library was performed following the 16S rRNA gene Metagenomic Sequencing Library Preparation Illumina protocol (Part #15044223 Rev. A). The gene-specific primer sequences used in this protocol were selected from [28] Illumina_16S_341F (TCGTCGGCAGCGTCAGATGTGTATAAGAGACAGCCTACGGGNGGCWGCAG) and Illumina_16S_805R (GTCTCGTGGGCTCGGAGATGTGTATAAGAGACAGGACTACHVGGGTATCTAATCC) which target the 16S V3 and V4 region, resulting in a single amplicon of approximately 460 bp. Overhang adapter sequences were used together with the primer pair sequences for compatibility with Illumina index and sequencing adapters. After 16S rRNA gene amplification, the DNA was sequenced on a MiSeq Sequencer according to the manufacturer's instructions (Illumina) using the $2 \times 300$ bp paired-end protocol.

### 2.8. 16S rRNA Gene Data Analysis

Raw sequence reads were processed using the pipeline implemented by the dada2 R library [29]. Briefly, the following steps were performed: low-quality reads were filtered, and end-trimming was performed. Reads were then merged and paired to obtain the full denoised sequences, and chimeras were removed. Following this, an inference of sequence variants (true diversity) for each sample was performed to obtain the amplicon sequence variant (ASV) table. After the removal of singleton reads from the dataset, 2,686,901 sequences remained, with an average of 34,447 sequences per sample. Taxonomic information was assigned to each variant based on the SILVA database (Silva Project's version 132 release). The Naive Bayesian classifier was used to assign up to genus level and 97% identity blast matching to species level.

### 2.9. Statistics

Jamovi (jamovi version 1.0.0, 2019, www.jamovi.org, accessed on 10 January 2019) was used for statistical analysis. GraphPad Prism version 5 (GraphPad Software Inc., San Diego, CA, USA) was used to create the figures. The distributions of data were assessed using the Shapiro–Wilk test; non-parametric tests were used where data were not normally distributed. A one-factor repeated-measures ANOVA was used to assess the main effect of the condition (pre-probiotics, post-probiotics, pre-placebo, and post-placebo) on salivary-pH, nitric oxide metabolites, and BP. Post-hoc analysis was conducted following a significant main effect using paired samples t-tests with Tukey's correction for multiple pairwise comparisons. The alpha level for declaring statistical significance was set at

$p \leq 0.05$. Data are presented as mean $\pm$ standard deviation (SD) unless otherwise stated. Probability values are expressed with 95% confidence intervals (95%-CI) where appropriate.

## 3. Results

### 3.1. Impact of 14 Days of Streptococcus salivarius M18 on the Oral Microbiome

The 16S rRNA data were analysed for abundance changes in the community composition. The low-abundance bacterial genus *Oribacterium* showed significant differences between visits (pre-probiotic 0.028%, post-probiotic 0.025%, pre-placebo 0.018%, post-placebo 0.027%, $p = 0.045$). Species-level analysis showed that these differences lay in *Oribacterium asaccharolyticum* ACB7 (pre-probiotic 0.0022%, post-probiotic 0.0016%, pre-placebo 0.0008%, and post-placebo 0.0015%, $p = 0.015$), and *Oribacterium sinus* (pre-probiotic 0.016%, post-probiotic 0.015%, pre-placebo 0.011%, and post-placebo 0.017%, $p = 0.033$ respectively). Post-hoc tests revealed that the bacteria varied between visits. At the genus level, the relative abundance of this bacteria was higher in the pre-probiotics measurement than the pre-placebo measurement (0.028% and 0.018%, respectively, $p = 0.019$) and increased from pre-placebo to post-placebo (post-placebo—0.027%, $p = 0.008$). The low-abundance species *Oribaterium sinus* followed a similar pattern (pre-probiotics 0.02%, pre-placebo 0.011%, $p = 0.011$, and post-probiotics 0.018%, $p = 0.007$). *Oribaterium asaccharolyticum* ACB7 was also higher in pre-probiotics than pre-placebo (pre-probiotics 0.002%, pre-placebo 0.0008%, $p < 0.001$). This species was also higher in the post-placebo measurement than in the pre-placebo measurement (post-placebo 0.002%, $p = 0.015$). No significant differences were found in the remainder of the oral bacteria following probiotic supplementation or placebo at genus or species level (all $p > 0.05$) (Figure 1). The Shannon diversity index and the number of observed OTUs also did not differ between study arms and did not change following supplementation (all $p > 0.05$).

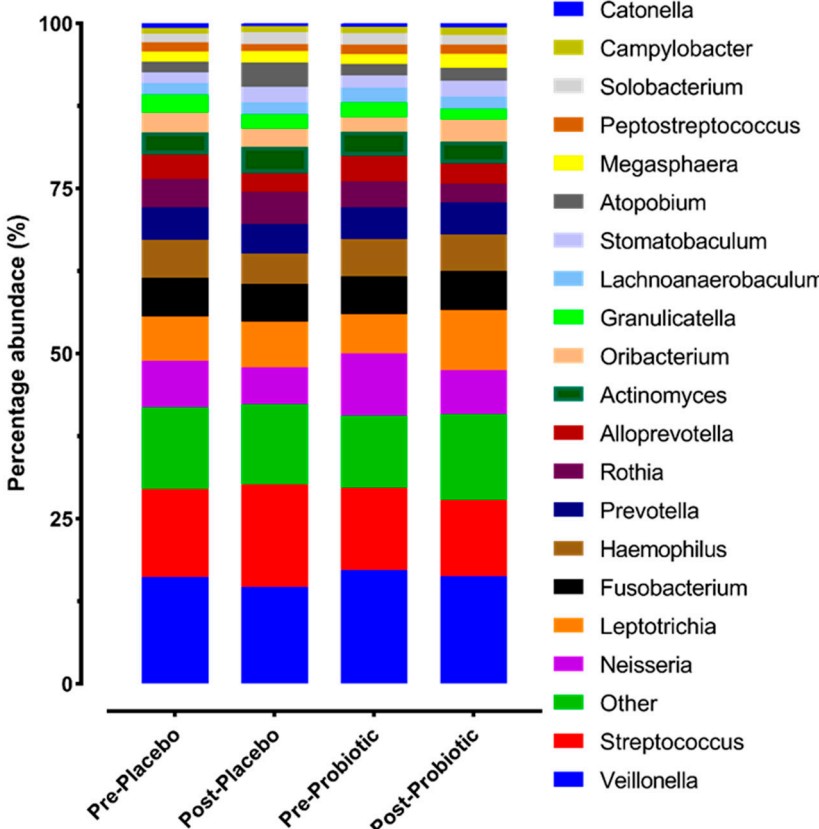

**Figure 1.** Genus level analysis of tongue swab samples (comparing the probiotic supplement intake with placebo intake). The 20 most abundant genera are shown. The green boxes with "other" contain the remaining genera grouped together.

### 3.2. Impact of Streptococcus salivarius M18 on Nitric Oxide Metabolites, Blood Pressure, and Salivary pH

#### 3.2.1. Nitric Oxide Metabolites

The levels of nitric oxide metabolites in the saliva and plasma are presented in Figure 2. Plasma nitrite increased following probiotic supplementation ($p = 0.003$). In the probiotics arm, plasma nitrite increased from baseline ($p = 0.001$, 95% CI 192–250 nM) and was higher following probiotics than at all other time points (pre-placebo $p < 0.001$, 95% CI 133–191 nM and post-placebo $p = 0.001$, 95% CI 142–199 nM). There was no change in the placebo arm ($p = 1.0$), and there were no differences between baseline measurements ($p > 0.05$). Plasma nitrate, salivary nitrite, and salivary nitrate did not change significantly at any time point (all $p > 0.05$).

#### 3.2.2. Blood Pressure and Salivary pH

There were no significant differences in SBP, DBP, MAP, or salivary pH following probiotic supplementation or placebo, and there were no differences in baseline measurements of these variables (all $p > 0.05$, Figure 2).

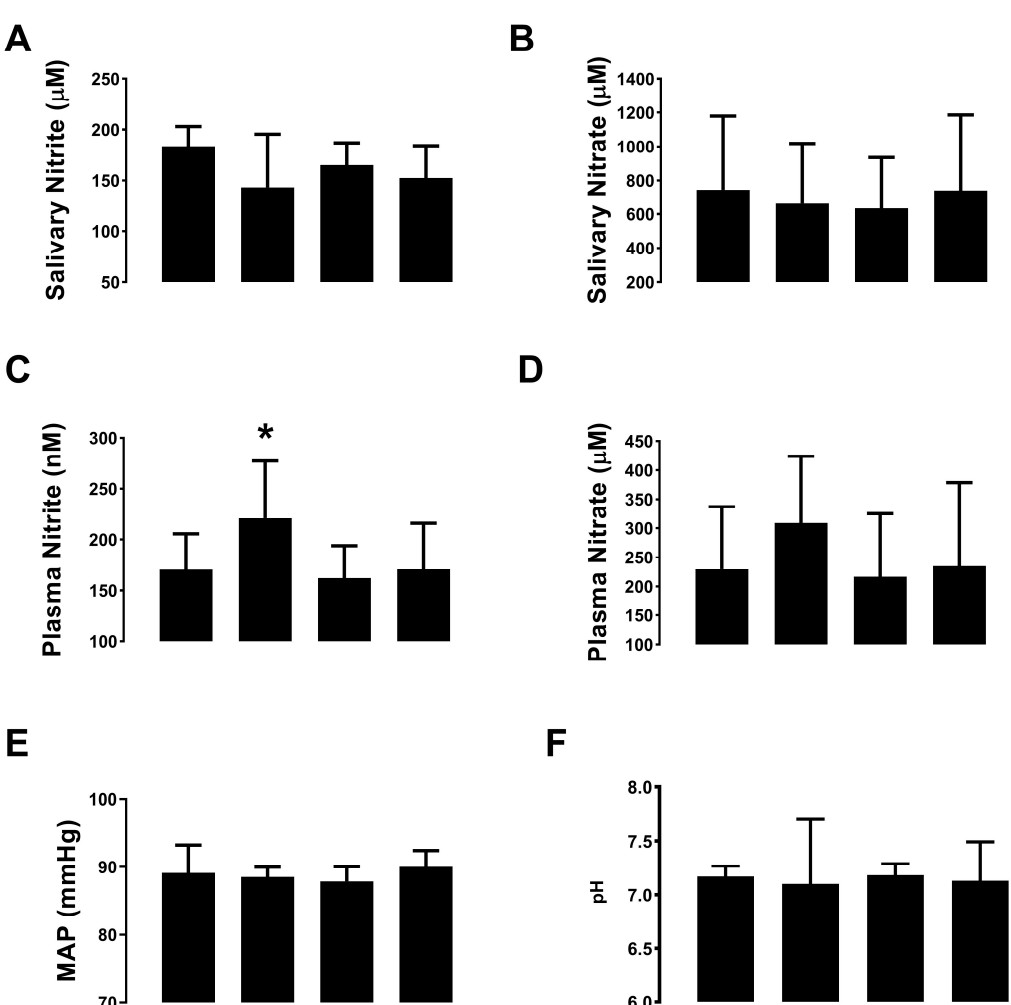

**Figure 2.** *Cont.*

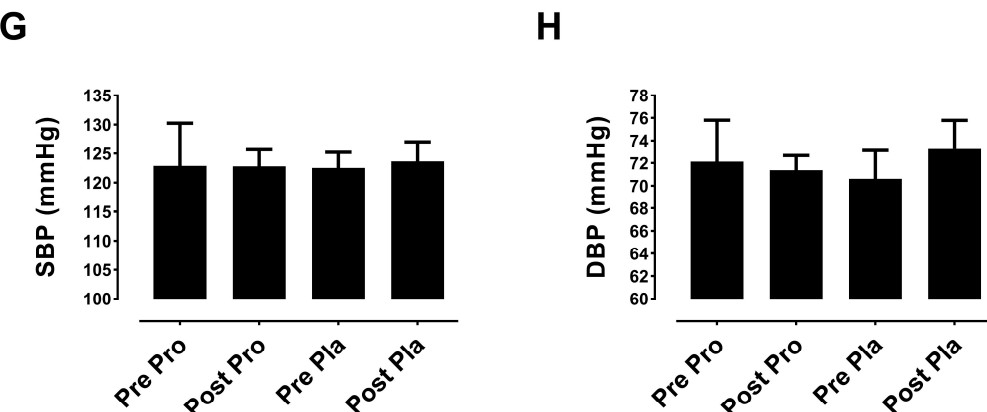

**Figure 2.** Salivary (nitrite) (**A**), salivary (nitrate) (**B**), plasma (nitrite) (**C**), plasma (nitrate) (**D**), MAP (**E**), salivary pH (**F**), SBP (**G**), and DBP (**H**). Pre Pro and Post Pro represent pre-probiotics and post-probiotics timepoints, respectively. Similarly, Pre Pla and Post Pla represent the pre- and post-placebo measurements. * denotes significant differences between timepoints.

## 4. Discussion

Despite the emerging importance of the oral microbiome and the nitrate to nitrite to nitric oxide pathway to human health, little information exists on how oral probiotic interventions may interact with this critical system. The data presented here show that supplementing for fourteen days with *Streptococcus salivarius* M18 increased plasma nitrite, the best-known approximation of nitric oxide bioavailability. Importantly, the increase in plasma nitrite is greater than the normal biological variation we have previously reported for this metric, indicating that *Streptococcus salivarius* M18 caused a biologically meaningful change in nitric oxide bioavailability [30]. This increase in plasma nitrite occurred without alterations to the abundance of bacteria thought important to nitric oxide generation. Despite the elevations in plasma nitrite following *Streptococcus salivarius* M18, blood pressure was not lowered in this group of healthy participants. The mechanism behind *Streptococcus salivarius* M18's effects on plasma nitrite requires clarification, and the potential of this probiotic to provide cardiovascular benefits in those with elevated BP needs further investigation. These mechanisms could include nitrite production by *Streptococcus salivarius* M18 [13] or metatranscriptomic changes in the community in the presence of this probiotic.

### 4.1. Interactions of Streptococcus salivarius M18 with the Host Microbiota

We observed changes in the abundance of *Oribacterium* at both species and genus levels. However, this bacteria was observed in low-abundance and appeared to be variable over the course of the four visits. Whilst *Oribacterium* has previously been associated with caries, it is not thought to be associated with nitrate or nitrite reduction [13,26,31]. Therefore, its change in abundance does not explain the observed increase in plasma nitrite. The lack of consistent changes to the oral microbiome following *Streptococcus salivarius* M18 is not wholly surprising as it has previously been demonstrated following next-generation sequencing survey analysis, that administration of *Streptococcus salivarius* M18 did not change the overall ecology of the oral microbiome [32]. Burton, Wescombe, et al. [11] used culturing methods to investigate the ability of *Streptococcus salivarius* M18 to persist in healthy individuals finding that higher doses promoted higher retention of M18. Interestingly, these authors [11] also found that *Streptococcus salivarius* M18 replaced indigenes *Streptococcus salivarius,* which may explain why the total abundance of *Streptococcus salivarius* did not increase in their data or why we did not see an increase in *Streptococcus* at species or genus level despite administering a high dose. Changes to the overall microbial community structure can be undesirable in healthy adults and are not necessarily required for *Streptococcus salivarius* M18 to confer benefits [32]. Bacteriocin and anti-inflammatory mechanisms may be the drivers behind the therapeutic effects of *Streptococcus salivarius* M18, or the probiotic strain may replace commensal species.

It is also important to note that we only performed bacterial abundance analysis; therefore, we cannot assess whether the resident bacteria adapted their nitrate/nitrite metabolism on a metatranscriptomic level in response to the presence of the probiotic. Upregulation of nitrate/nitrite metabolism gene expression by the resident microbial community could explain the increased appearance of nitrite in the plasma via improved utilisation of nitrate from the habitual diet [23]. Alternatively, the probiotic itself may have contributed to the increase in plasma nitrite. Streptococcus spp., including strains of *S. salivarius*, are known nitrite-producing species, meaning that they can produce nitrite by nitrate reduction or other metabolic pathways [14]. Additionally, there have been previous reports of *Streptococcus salivarius* reducing nitrite [13]. Direct reduction of nitrite in the oral cavity has yet to be confirmed as a contributor to the appearance of nitrite in the plasma, and culturing of *Streptococcus salivarius* M18 would be required to confirm its nitrate/nitrite reduction capabilities. The slow time course of nitrite reduction has previously been the reason for discounting nitrite reducers as important to the human nitrogen cycle [33]. However, recent data from [34] suggested that nitrite reducers may, in fact, be important as the nitric oxide they generate may diffuse through the vasculature of the tongue. Experiments investigating the speed of nitrite reduction in the oral cavity are required to confirm the importance of nitrite reducers to the nitrate–nitrite–nitric oxide pathway.

### 4.2. Interactions of Streptococcus salivarius M18 with Plasma and Saliva

Whilst speculative, a further possible explanation for the increase in plasma nitrite may be that endogenous nitric oxide synthesis was upregulated via a host immune response to the large amounts of probiotic bacteria administered. Probiotics can modulate macrophage function, and their positive effects are related to their ability to balance pro-inflammatory and anti-inflammatory cytokines [35,36]. In addition, it has long been known that nitric oxide is an anti-microbial effector molecule produced by macrophages and capable of providing immune protection at low physiological levels via regulatory and effector functions [37]. *Streptococcus salivarius* M18 and its close relative *Streptococcus salivarius* K12 are both known to downregulate the expression of the pro-inflammatory cytokines IL-6 and IL-8 and have been shown to improve disease indices in inflammatory conditions such as gingivitis and periodontitis [10].

*Streptococcus salivarius* can also induce high levels of the anti-inflammatory cytokine interleukine-10 (IL-10) and low levels of pro-inflammatory IL-12 [12]. IL-12 is thought to play an integral role in innate and adaptive immunity by facilitating crosstalk between these systems [38]. Interestingly, S. thermophilus, which is closely related to *Streptococcus salivarius*, has been shown to increase levels of IL-12, providing protection against inflammation [38]. Furthermore, anti-inflammatory IL-2 and IL-12 have been shown to work synergistically, and IL-2-mediated immune activation was shown by Hibbs et al. [39] to increase nitric oxide (assessed via increased serum nitrate). Despite the possible connection between the anti-inflammatory properties of probiotics and those of nitric oxide, there has been little investigation of the effects of probiotics on nitric oxide production and circulating levels of nitrite and nitrate.

Korhonen et al. [40] investigated the possibility that nitric oxide synthesis was involved in the cellular actions of the gut probiotic *Lactobacillus rhamnosus* GG in J774 macrophages and human T84 intestinal epithelial cells. *Lactobacillus rhamnosus* GG has protective effects in the gut. However, its mechanism of action remained unclear. Korhonen and colleagues carried out an in vitro experiment showing that *Lactobacillus rhamnosus* GG induced low-level nitric oxide production (measured as nitrite accumulation) and that NOS inhibitors interrupted this process. They concluded that *Lactobacillus rhamnosus* GG induces nitric oxide production via the iNOS pathway and that nitric oxide production may contribute to the protective actions of this probiotic. Further interrogation of the samples collected in this study would be required to provide mechanistic evidence that the observed increase in plasma nitrite is indeed related to a host immune response.

The timing of probiotic intake was 12 h before the subsequent measurement, which may explain the lack of changes in salivary measurements. This time period was chosen to allow residual probiotics to clear from the oral cavity to allow detection of adherence of the probiotic strain to the tongue papillae and thus assess any true displacement of bacteria involved in nitric oxide generation. Species and strain level identification of *Streptococcus* spp. is not recommended via 16S rRNA gene sequencing analysis at 300 bp as the resolution is not specific enough to determine nucleotide differences, meaning that we could not directly assess the persistence of *Streptococcus salivarius* M18 from the 16S data. However, Burton and colleagues [11] have previously identified that *S. salivarius* M18 can replace indigenous *S. salivaruis* without affecting overall numbers, indicating that adherence to the probiotic strain can be inferred from changes in *Streptococcus salivarius*. Whilst we did not see significant differences in *Streptococcus salivarius*, further examination of samples using culturing methods could be performed to confirm whether adherence occurred.

### 4.3. Effect of Streptococcus salivarius M18 on Blood Pressure

Although we observed an increase in plasma nitrite, we did not observe changes in blood pressure. It is suggested that reductions in BP following probiotic administration are more prevalent where baseline BP is elevated [41]. The recent review by et al. [42] performed subgroup analysis highlighting that the hypotensive effect of probiotics is more likely to occur when baseline BP is >130/85 mmHg. Our participants were healthy, with baseline BP substantially below this value.

*S. thermophilus* is often reported to reduce blood pressure when administered either alone or in combination with other bacterial strains. The exact mechanisms by which these changes occur remain unclear, with most studies to date focussing on the ability of probiotics to improve the blood lipid profile. Ito et al. [43] administered *Streptococcus thermophilus* YIT 2001 to healthy and mildly hyper-LDL-cholesterolaemic adults showing that the strong anti-oxidative activity of this probiotic improved risk marker values of oxidative stress and reduced blood pressure. These effects were more pronounced in subjects with higher oxidative stress at baseline. Agerholm-Larsen et al. [44] found that one strain of *Enterococcus faecium* and two strains of *Streptococcus thermophilus* (CAUSIDO® culture) administered for eight weeks reduced LDL cholesterol and increased fibrinogen in overweight subjects, while the mechanisms remained unclear, the authors attributed increases in fibrinogen to immunostimulation. Kawase et al. [45] showed that *Lactobacillus casei* TMC0409 and *Streptococcus thermophilus* TMC 1543 decreased the atherogenic index (measured as total cholesterol—high-density lipoprotein cholesterol/high-density lipoprotein cholesterol) and systolic blood pressure. Similarly to the findings of Agerholm-Larsen et al. [44], there was no precise mechanism identified for these positive changes.

Khalesi Saman et al. [42] also found that hypotensive effects may be more significant when the duration of the probiotic intervention is ≥8 weeks. The period of intake in this study was 14 days, significantly shorter than the eight weeks recommended for BP reduction. However, *Streptococcus salivarius* M18 was shown to persist in the oral cavity at its highest at 14 days of administration [11]. As our main aim was to capture the period in which the introduction of a probiotic may have the most impact on the oral microbiome and the nitrate–nitrite–nitric oxide pathway, we tailored our administration period based on Burton and colleagues' observations [11].

### 4.4. Limitations

The BP-lowering effects of gut probiotics have mostly been attributed to improvements in blood lipid chemistry, reductions in serum leptin, and oxidative stress [2–6]. The data shown here suggest that probiotics designed to act in the oral cavity may also confer systemic benefits, perhaps through the generation of nitric oxide, upregulation of NOS, or reduction of oxidative stress. As we are limited in our ability to provide mechanistic evidence for the observed increase in plasma nitrite, these possibilities remain speculative

until further data are collected. Furthermore, our pilot study was underpowered to detect reductions in BP, and the sample was limited to healthy males; future work should be conducted in larger and heterogeneous groups. Analysis of bacterial activity may clarify whether the elevated plasma nitrite was due to an increase in the microbial generation of nitric oxide. Additionally, collecting markers of immune function and anti-oxidative capacity alongside assessment of nitric oxide metabolites following *Streptococcus salivarius* M18 consumption may further clarify the mechanisms.

**5. Conclusions**

Consuming the oral probiotic *Streptococcus salivarius* M18 for fourteen days increases levels of nitrite in the blood plasma. The plasma nitrite increase did not appear to be associated with changes in the abundance of bacteria considered important to nitric oxide generation. Further research is needed to confirm, but the data suggest that the effects of *Streptococcus salivarius* M18 on nitric oxide bioavailability may be related to the production of nitric oxide through alternative mechanisms such as nitrite production, direct generation of nitric oxide in the oral cavity, upregulation of iNOS, or reduction of oxidative stress. In addition, elevated plasma nitrite did not translate into lower BP in this group of healthy male volunteers. However, further investigation of the systemic effects of *Streptococcus salivarius* M18 is warranted, given the proposed therapeutic benefits of increased nitric oxide bioavailability and the strong links between oral health and cardiovascular risk.

**Supplementary Materials:** The following supporting information can be downloaded at: https://www.mdpi.com/article/10.3390/applmicrobiol3030054/s1, File S1: Medical Questionnaire for Physiological Testing; File S2: Inclusion/Exclusion criteria; File S3: CONSORT Checklist.

**Author Contributions:** Conceptualization, M.C.B. and C.E.; methodology, M.C.B. and C.E.; formal analysis, M.C.B. and B.T.R.; writing—original draft preparation, M.C.B.; writing—review and editing, C.E., F.H., N.S., B.T.R. and A.S.; visualization, A.S.; project administration, M.C.B.; funding acquisition, C.E. All authors have read and agreed to the published version of the manuscript.

**Funding:** The research was supported by funding from the Hannah Dairy Research Foundation.

**Institutional Review Board Statement:** This research was approved by the School of Health and Life Sciences Ethics Committee at the University of the West of Scotland—approval number 2018-5000-3633—date May 2018. All procedures described were conducted in accordance with the Declaration of Helsinki 1974 and its later amendments.

**Informed Consent Statement:** Not applicable.

**Data Availability Statement:** Data is available upon request to the corresponding author.

**Conflicts of Interest:** The authors declare no conflict of interest.

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
