# Peer review of "The Probiotic Streptococcus salivarius M18 Increases Plasma Nitrite but Does Not Alter Blood Pressure: A Pilot Randomised Controlled Trial"

_2673-8007, doi:10.3390/applmicrobiol3030054_

Round 1

Reviewer 1 Report

Dear authors,

This is an interesting study but the methodology should be entirely. Please consider my comments:

Materials and methods

1.     Report the data according to EQUATOR and the checklist in supplementary file

2.     Section 2.1 Add the number and the date of the approval by the ethical committee 

3.     "Health status was confirmed by completion of a medical questionnaire" Please give the reference of this questionnaire

4.     "The World Health Organisation’s oral health questionnaire was used to ascertain oral health status" Add one ref

5.     How did the sample size determinate?

6.     What did the participant was fasted and euhydrated state after consuming 500 ml of water 1 h 125 before each trial?

7.     Precise inclusion/exclusion criteria? Date of the study? Where did the study realized?

Reviewer 2 Report

With interest I’ve read the paper “The probiotic Streptococcus salivarius M18 increases plasma

nitrite but does not alter blood pressure”.

The authors sought to determine the effects of Streptococcus salivarius M18 supplementation on plasma and salivary nitrate and nitrite levels and blood pressure by conducting a cross-over study with 10 healthy males. They found that Streptococcus salivarius M18 supplementation increased plasma nitrite but did not lower BP in healthy normotensive participants and did not change the community of bacteria important to NO generation. The study thoroughly presented, but the main limitation is a small sample size and a short time of probiotic intake. The study is of interest to a wide range of readers among dental professionals. However, some comments should be addressed.

Why the sample size was just 10 participants? Can you provide power analysis for your results?

Please, provide more details on the exclusion and inclusion criteria. What do you meant by “good cardiovascular and oral health“?

Please, explain, why the period of probiotic intake was set at 14 days? This should be discussed in the discussion section and the references should be provided on the influence of the period of probiotic intake on its effects.

Round 2

Reviewer 1 Report

Thank you for considering my comments